# Expected Scopes of Health Emergency and Disaster Risk Management (Health EDRM): Report on the Expert Workshop at the Annual Conference for the Japanese Association for Disaster Medicine 2020

**DOI:** 10.3390/ijerph18094447

**Published:** 2021-04-22

**Authors:** Shuhei Nomura, Ryoma Kayano, Shinichi Egawa, Nahoko Harada, Yuichi Koido

**Affiliations:** 1Department of Health Policy and Management, School of Medicine, Keio University, 35 Shinanomachi, Shinjuku-ku, Tokyo 160-8582, Japan; s-nomura@keio.jp; 2Department of Global Health Policy, Graduate School of Medicine, The University of Tokyo, 7-3-1 Hongo, Bunkyo-ku, Tokyo 113-0033, Japan; 3World Health Organization Centre for Health Development, 1-5-1 Wakinohama-kaigandori, Chuo-ku, Kobe 651-0073, Japan; 4International Cooperation for Disaster Medicine Laboratory, International Research Institute of Disaster Science (IRIDeS), Tohoku University, 468-1, Aramaki-aza-Aoba, Aoba-ku, Sendai 980-8572, Japan; egawas@surg.med.tohoku.ac.jp; 5Department of Mental Health and Psychiatric Nursing, School of Nursing, Faculty of Medicine, University of Miyazaki, 5200 Kiyotakecho Kihara, Miyazaki, Miyazaki 889-1692, Japan; nahoko-harada@umin.ac.jp; 6Japan Disaster Medical Assistant Team Secretariat, National Hospital Organization Headquarter, 3256 Midoricho, Tachikawa, Tokyo 190-0014, Japan; koido@outlook.jp

**Keywords:** health emergency and disaster risk management (Health EDRM), health EDRM research network, health EDRM knowledge hub, health EDRM research agenda, WHO thematic platform for health EDRM

## Abstract

The World Health Organization (WHO) and its partners established the WHO Thematic Platform for Health Emergency and Disaster Risk Management Research Network (HEALTH EDRM RN) in 2016 to respond to the increasing burden of recent health emergencies and disasters. The mission of the HEALTH EDRM RN, whose secretariat is located at the WHO Kobe Centre (WKC), is to promote global research collaboration and strengthen research activities to inform policies and programs by generating new evidence to manage health risks associated with all types of emergencies and disasters. With the strong support and involvement of all WHO regional offices, the HEALTH EDRM RN now works with more than 200 global experts and partners to pursue its mission. The first Core Group Meetings of the HEALTH EDRM RN were held on 17–18 October 2019, and concluded with the HEALTH EDRM RN-activity priorities to (1) promote operational research to better meet the needs of emergency- and disaster-exposed individuals and communities and efforts to translate science to policies and programs and (2) strengthen the research capacity of the Health EDRM community. In collaboration with the Japanese Association for Disaster Medicine, the WKC held a workshop on 21 February 2020, in which 20 Japanese experts from different research fields participated to further discuss these two points. This paper summarizes the discussion at the workshop.

## 1. Introduction

At the third United Nations World Conference on Disaster Risk Reduction (WCDRR), held in Japan in 2015, human lives, health, and livelihoods were included in the Sendai Framework for Disaster Risk Reduction 2015–2030 (Sendai Framework) as the main objectives of disaster risk reduction [1]. As a key goal of disaster risk reduction, the broad intersection of health and disaster risk reduction is captured in the concept of Health Emergency and Disaster Risk Management (Health EDRM). Health EDRM refers to the systematic analysis and management of health risks posed by health emergencies and disasters and plays an important role in preventing and mitigating hazards, exposure, and vulnerability along with enhancing coping capacity in terms of preparedness, response, and recovery [2]. Thus, it encompasses various disciplines, including emergency and disaster medicine, disaster risk reduction, humanitarian response, community health resilience, and health system resilience [3].

In March 2016, the first international conference on the health aspects of the Sendai Framework was jointly organized by the United Nations Office for Disaster Risk Reduction and the World Health Organization (WHO) [4]. The conference promoted the Bangkok Principles, identifying seven areas for mainstreaming disaster risk reduction within a health context, including fostering cross-sectoral transboundary collaboration for all hazards. The WHO and its partners also established the WHO Thematic Platform for Health Emergency and Disaster Risk Management Research Network (HEALTH EDRM RN) in the same year as a subgroup of the Thematic Platform for Health EDRM, with the WHO Centre for Health Development (also known as WHO Kobe Centre: WKC; Kobe, Japan) as secretariat. With more than 200 global experts and partner institutions as of 2020, the major missions of the HEALTH EDRM RN include the promotion of global research collaboration among stakeholders, such as academia, government agencies, and the private sector; strengthening research activities that generate evidence needed to manage health risks associated with all types of health emergencies and disasters; and informing policies and practices more effectively [5,6].

The expert meeting at the 2018 Asia Pacific Conference for Disaster Medicine in Japan was organized by the WKC and convened 32 experts from 12 countries, including the members of the HEALTH EDRM RN (namely, the 2018 Kobe Expert Meeting). As the outcome of the meeting, key research themes and challenges for Health EDRM were identified: health data management, psychosocial management, community risk management, health workforce development, and research methods and ethics. Details of these themes and challenges are described elsewhere [7]. In October 2019, the first HEALTH EDRM RN Core Group meeting was held on 17–18 October 2019, with WHO representatives (from headquarters and six regional offices) as well as external experts from Public Health England, Chinese University of Hong Kong, Japanese Association for Disaster Medicine, Japan Society of Disaster Nursing, World Association for Disaster and Emergency Medicine, etc. At the meeting, the expected scope of HEALTH EDRM RN activities was discussed (Table 1), and the meeting concluded with the HEALTH EDRM RN activity priorities to promote operational research to better meet the needs of emergency- and disaster-exposed individuals and communities, to promote efforts to translate science to policies and programs, and to strengthen the research capacity of the Health EDRM community, such as quality of the research environment, availability of appropriate support and guidance, research career development, etc. [8].

## 2. Materials and Methods

In this context, the WKC, in collaboration with the Japanese Association for Disaster Medicine, held a workshop on 21 February 2020, in which 20 Japanese experts from different research fields participated. The workshop was comprised of 11 presentations and a panel discussion with six experts. The objectives of the presentations were to discuss (a) the design and implementation of operational research and its translation into policy and practice in the context of Health EDRM and (b) progress and challenges in HEALTH EDRM RN activities to strengthen the research capacity of the Health EDRM community. The first six experts gave presentations about objective (a) from different perspectives, all experts exchanged their opinions, and then the moderator summarized the opinions. This workshop was open to the public, with an audience of about 100 people, and their opinions were also taken into account. Similarly, for objective (b), five experts gave presentations, and after exchanging opinions, the moderator summarized the results. The panel discussion was held with an objective to highlight (c) research areas that the HEALTH EDRM RN is expected to focus on as priority areas. At the end of the discussion, the moderator summarized the opinions. This paper summarizes the discussions at the workshop.

## 3. Results

### 3.1. Design and Implementation of Operational Research and Its Translation into Policy and Practice

The interaction of science, policy, and practice is traditionally scarce and needs to be improved [4,9]. The participants shared and discussed their experience with operational research and its translation into policy and practice. Operational research is described by Remme et al., (2010) as research aimed at solving current operational problems in specific health programs and in specific health system service delivery locations (hospitals, etc.) [10]. The characteristic of operational research is that it is demand-driven and has a strong focus on problem solving and an urgent need to find solutions.

While the majority of operational research is concerned with quantitative methods (quantitative surveys, statistical and mathematical modelling or inferences, experiments, etc.), qualitative methods, such as interviews, focus groups, and the Delphi technique, are also being recognized for their contribution to contextual and narrative data needed for, for example, family planning and reproductive health services [11]. The workshop participants agreed that mixed methods combining quantitative and qualitative approaches are more suitable for operational research in Health EDRM. These methods enable researchers to design a single research study that answers questions about the complex nature of the phenomenon of disasters, which require various solution strategies, especially at the level of operation execution [12,13,14]. In recent years, there has been an increase in operational research in the healthcare setting using mixed methods [15], but few studies have applied it to Health EDRM research. The basic designs of the mixed methods research in Health EDRM include an analysis of numerical survey responses with Likert scales and subsequent thematic analysis of interview data and an iterative content analysis of focus group data followed by an analysis of binary responses [16].

Several reviews of operational research in disaster risk management have highlighted accurate data acquisition as an essential step in the implementation of operational research [17,18]. The workshop participants addressed how the lack of a standardized data-collection mechanism hampers operational research in health emergency and disaster settings. Data collection during a health emergency and disaster poses many challenges, including secure access to the affected area, preparing resources for the data collection, obtaining informed consent from the affected population, and fragmenting the data collection and reporting among different relief teams (often from around the world) [19]. All of these challenges contribute to a significant lack of scientific evidence in Health EDRM research in a systematic way. Operational research that is not based on adequate and accurate data as well as support activities based on this research may even have adverse effects on post-emergency/disaster reconstruction efforts [20], contrary to the do no harm principle—a core humanitarian principle propagated by Mary Anderson [21] to protect the beneficiaries of humanitarian assistance—which is considered the minimum standard to avoid causing inadvertent harm.

There is a growing body of research on the use of Geographic Information Systems (GIS) for operational research, which can help develop location analysis and obtain more accurate data on spatial elements (road networks, geographic obstacles, etc.) [22,23,24]. In addition, a successful case study was provided by the participants. In Japan, a timely surveillance of diseases and information sharing among the responding teams were difficult after the 2011 Great East Japan Earthquake that affected dozens of coastal communities along the shore of Japan’s Tohoku region. Based on this experience, the Joint Committee for Disaster Medical Record of Japan (which consisted of five organizations at the time, including the Japan Medical Association, Japanese Association for Disaster Medicine, Japan Hospital Association, Japan Society of Health Information Management, and Japanese Society for Emergency Medicine; it added the Japan International Cooperation Agency (JICA) and Japan Psychiatric Hospitals Association in 2018) developed Japan-Surveillance in Post Extreme Emergencies and Disasters (J-SPEED) [19]. J-SPEED is a standardized daily medical report system used by different emergency medical teams (EMTs) that come to the affected areas after the occurrence of health emergencies and disasters. J-SPEED was put into operational research on a large scale for the first time in the wake of the 2016 Kumamoto Earthquake, contributing to the disaster response headquarters’ understanding of the overview of health care needs. In 2017, the WHO proposed the Minimum Data Set (MDS), based on J-SPEED, as a standard data collection system for EMTs. In March 2019, MDS was first introduced in Cyclone Idai in Mozambique and is now being used in health emergencies and disasters around the world. The items of J-SPEED were updated in 2019 to correspond to the WHO’s international standard MDS.

The keys to success that we could learn from this case study included (1) organizations capable of reporting were identified in advance; (2) the organizations were provided a practical and standardized daily report format for field activities; and (3) there was a headquarters capable of linking the reported data to the emergency- and disaster-response.

### 3.2. Progresses and Challenges to Strengthen the Research Capacity of the Health EDRM Community

The participants discussed the progress and challenges of the HEALTH EDRM RN activities to strengthen the research capacity of the Health EDRM community. One of the five research themes identified at the 2018 Kobe Expert Meeting was research methods and ethics [7]. Further, the participants acknowledged that standardized methodological guidance for planning and conducting research on Health EDRM was urgently needed. Such guidance could promote high-quality research that supports the best evidence-based policies and actions and provide researchers with optimal methods for their research [25]. As a response, the WKC, in collaboration with the WHO headquarters and regional offices and representatives of the HEALTH EDRM RN, has begun to develop the WHO Guidance on Research Methods for Health-EDRM. This guidance consists of 6 sections and 43 chapters that address a wide range of research fields, and more than 160 global experts contributed to its development (https://extranet.who.int/kobe_centre/en/project-details/GUIDANCE_ResearchMethods_HealthEDRM, accessed on 31 March 2021). This is the first comprehensive WHO guide for research methods in the area of health EDRM. The workshop participants shared their expectations of guidance to strengthen the research capacity of the Health EDRM community. Communicating this guidance as well as the background behind it to a wide range of people, including not necessarily only Health EDRM researchers, but also policy-makers, administrative staff, and local health workers, was encouraged for better mutual understanding and collaboration [26].

In addition, the participants acknowledged that access to the data needed to conduct operational research is a major challenge for Health EDRM. Data collected by relief agencies are often not reported in a format that is appropriate for scientific decision making and analytical purposes [27]. By taking into account the do no harm principle [21], it is inevitable that the data will be incomplete or not measured correctly. Data are often collected for reporting and accountability and are often not suitable for research. Therefore, Health EDRM researchers often have to collect the data themselves. However, the unpredictable nature of emergencies and disasters as well as ethical and humanitarian reasons make it difficult to collect data during emergencies and disasters.

In relevance to the above, the participants addressed the limited career prospects for young scientists in the Health EDRM community as well as a resulting chronic shortage of qualified researchers in this field as challenges to strengthen the research capacity on the community. Being a researcher is a competitive job because of the high standards for the obtainment of tenure and grants. Today, much of this competition is often assessed based on a scientist’s number of publications, the number of times they are cited, the impact factors of the journals they publish in, and their h-index. These indices have great consequences. In some countries and disciplines, publication in a journal with an impact factor lower than 5.0 is officially considered to have no value [28]. However, despite the significant research momentum to generate evidence leading to disaster-risk management in response to the increasing frequency and severity of health emergencies and disasters around the world [29,30], the impact factors of academic journals focusing on disaster-risk reduction and related areas are lower than that of other major health and medical fields [31]. There is a great amount of literature and discussion criticizing these indices and their use [28,32], such as the San Francisco Declaration on Research Evaluation (DORA) [33]; however, they still influence careers greatly. As a result, it has become important, especially for young researchers, to consider the implications of these indicators and the consequences of their use. For these reasons, researchers can be discouraged from pursuing this field [34].

### 3.3. Priority Research Areas

Operational research on Health EDRM is expected to address a wide range of hazards and events, including those associated with natural, man-made, and other complex emergencies. During the panel discussion among the workshop participants, the complex health hazards associated with different emergencies were addressed, and the following research areas were highlighted as areas on which the HEALTH EDRM RN is expected to focus.

#### 3.3.1. Mental Health Including Dementia and Other Cognitive Disabilities

Emergencies and disasters can have short-term, medium-term, and long-term effects on people’s mental health [35]. They can also worsen pre-existing psychological conditions, including substance use problems. The participants emphasized the importance of operational research from the perspective of preventive care, such as identifying affected individuals who do not meet the diagnostic criteria for mental illness but are potentially at risk and preventing them from developing the illness in the medium- to long-term after the emergency or disaster [36]. In addition, people living with and affected by dementia are particularly in need of assistance when faced with health emergencies and disasters. Inadequate access to specific services needed by persons with disabilities (such as rehabilitation, assistive devices, access to social workers and interpreters, etc.) further hampers access to basic mainstream support such as water, shelter, food, and health care [37]. There is a clear lack of data and research on the magnitude of the impact of emergencies and disasters on dementia patients and the issues surrounding them. Operational research is needed to expand the evidence base to explore the experiences and protective needs of people with dementia and other cognitive disabilities in emergencies and disasters around the world.

#### 3.3.2. Business Continuity Planning

Some literature pointed out that despite its importance, there is a significant lack of operational research in the area of business continuity [17,18]. This knowledge gap was particularly highlighted after the 11 September attacks on the World Trade Center [17]. In addition to responding to rapidly increasing healthcare demands, the business continuity plan of healthcare must focus on the safety of staff and buildings, the continuous operation of critical infrastructure (such as communications, power generation, water, and sanitation services), and the maintenance of medical devices, equipment, supplies, utilities, and consumables [38,39]. Operational research is needed to evaluate the degree of the loss of function of healthcare systems (e.g., service delivery, financing, health workforce, and other inputs) in the event of an emergency or disaster [40,41], as well as the level of preparedness [42], to improve the resilience of healthcare facilities [43].

#### 3.3.3. Malnutrition—Both Overnutrition and Undernutrition

Emergencies and disasters can result in severe food shortages that can seriously affect the nutritional status of the affected population. This often leads to severe protein and energy malnutrition and micronutrient deficiencies, greatly affecting the health consequences of the affected population [44]. Furthermore, even if food stockpiles and relief supplies reach the affected population, they are not necessarily well-balanced in nutrition and tend to be diets high in proteins and carbohydrates. For example, in the 2011 Great East Japan Earthquake, it was difficult to distribute fresh vegetables, meat, fish, and dairy products containing a good balance of proteins and carbohydrates to the affected population in shelters [45]. Even one month after the earthquake, the diet of many of people in shelters was mostly composed of highly processed food containing a high percentage of proteins and carbohydrates [46]. Operational research is urgently needed to assess the magnitude and impact of malnutrition (both overnutrition and undernutrition) associated with emergencies and disasters and to find solutions to logistical challenges to deliver a more balanced diet to the affected populations.

#### 3.3.4. Welfare and Nursing Care

The concept of Health EDRM is not only about minimizing the direct loss of human lives and health, but also involves aspects of welfare and nursing care, such as risk management for issues related to social well-being and livelihood security, including economic loss and accompanying poverty of the affected population. It also includes the maintenance of nursing care services, which are essential for the elderly and physically and mentally disabled persons. In order to promote such risk management in the welfare field in times of emergencies and disasters, it is necessary to integrate various local resources in the formal sector (nursing care, welfare, and other social services) and the informal sector (activities of volunteers and non-government organizations and mutual support among local residents). By developing a comprehensive system in normal times, necessary services can be provided in an integrated manner, such as medical care, nursing care, prevention, housing, and life support [47].

In Japan, the construction of the Community-Based Integrated Care System is progressing [48]. This system provides elderly people with medical and nursing care services and welfare services, including disability support, in a unified manner to allow them to continue living in their current communities [49]. There is an expectation that support for people requiring nursing care will not be interrupted even in times of emergencies and disasters. On the other hand, there has been little operational research to identify the know-how and issues for smooth collaboration between various local resources in times of emergencies and disasters [50].

#### 3.3.5. Security and Safety of Response Teams and Volunteers and Local Responders

In the event of a health emergency and disaster, multidisciplinary and multiorganizational response teams and volunteers work to provide various support in the affected areas in close liaison with local responders such as local authorities, health systems, and other bodies. However, compliance with codes of conduct and understanding of support systems for safe response activities [51], including emergency responses (e.g., search and rescue) and recovery efforts, is not sufficient. With regard to the protracted conflict and the expansion of activities by various armed groups, the intensification of emergencies and disasters, and the spread of emerging infectious diseases, security and safety management are required for those carrying out response activities [52,53]. Attention to the security and safety of response teams and volunteers as well as local responders became particularly pronounced after the 2001 terrorist attacks at the World Trade Center. Many informal, initial volunteers arriving at the scene to support a search and rescue operation were overwhelmed by the tragic scene confronting them [54]. Without adequate relevant training, skills, or experience, unaffiliated volunteers, also known as spontaneous volunteers, who offer to help or self-deploy to assist in emergency situations, likely become traumatized and themselves become victims of the emergencies and disasters [55]. Similarly, a mental-health impact on response teams, volunteers, and local responders was reported for the 2011 Great East Japan Earthquake and the subsequent Fukushima nuclear power plant accident in Japan [56,57,58,59,60], in which many of the workshop participants actually engaged. Operational research is urgently needed to assess the security and safety of response teams, volunteers, and local responders who work in difficult circumstances as well as the physical- and mental-health issues they suffer.

## 4. Summary and Conclusions

The discussion among the experts at the workshop was thought provoking, with participants agreeing that the mixed methods combining quantitative and qualitative approaches is more appropriate for operational research to better meet the needs of emergency- and disaster-exposed individuals and communities, and that comprehensive and appropriate data collection using standardized data collection mechanisms, such as J-SPEED and MDS, are required to adhere to the do no harm principle. Furthermore, active educational use of the WHO Guidance on Research Methods for Health-EDRM is an expected tool to strengthen research capacity in the Health EDRM community. Meanwhile, it was affirmed that career development is relatively difficult for young researchers in this field in today’s impact factor-oriented academic world. Finally, five areas were identified as priorities for operational research on Health EDRM: mental health, business continuity planning, malnutrition, welfare, and security and safety of responders and volunteers.

The objectives of the workshop were achieved through the participation of a large number of experts, the collection of various inputs and advice on developing operational research, and strengthening the research capacity of the Health EDRM community. The impact of the new coronavirus disease (COVID-19 pandemic) on the five areas highlighted in this workshop is significant, supporting the relevance of priority focus of operational research on these areas. Unlike other hazards, a pandemic has limited prospect of damage, and research in this area is both an urgent need and will contribute to future pandemic responses. The WHO Guidance on Research Methods for Health-EDRM will be updated to reflect new and important scientific evidence. In particular, given the current situation of the COVID-19 pandemic, the WKC and HEALTH EDRM RN plan to update the content in 2021 and add new chapters on the findings highlighted during this workshop.

## Figures and Tables

**Table 1 ijerph-18-04447-t001:** Recommendations for HEALTH EDRM RN activities to ensure their success and effectiveness [8].

Recommended Activities	Rationale
Promotion of operational research that reflects the needs of emergency- and disaster-exposed individuals and communities	Given that the interaction between science, policy, and practice is traditionally poor and in need of improvement, HEALTH EDRM RN activities are expected to address the development of mechanisms to facilitate operational research and the translation of research findings into policy and practice. By addressing these mechanisms, the HEALTH EDRM RN can provide a bridge for both researchers and policy-makers as well as practitioners for better mutual understanding and collaboration.
Promotion of translation of research findings into policy and practice	
Strengthening research capacity of the Health EDRM community	In order to maximize the impact of Health EDRM research and promote further development of individual research areas, the research capacity of researchers and research communities, including new researchers, experienced researchers, and research supervisors, need to be improved.

HEALTH EDRM RN: Health Emergency and Disaster Risk Management Research Network.

## Data Availability

Not applicable.

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
