# Peer review of "Expected Scopes of Health Emergency and Disaster Risk Management (Health EDRM): Report on the Expert Workshop at the Annual Conference for the Japanese Association for Disaster Medicine 2020"

_ijerph, 2021, doi:10.3390/ijerph18094447_

Round 1

Reviewer 1 Report

REVIEW  - "Expected Scopes of the Health Emergency and Disaster Risk Management (Health EDRM): Report of the Expert Workshop at Annual Conference for Japanese Association of Disaster Medicine 2020"

It is a very good conference report that is showing its main concern of organize and develop a broad and efficient future system to enhance technical proceedings into very complex and harmful situations. This report is well structured and very clear. Only one very, very simple observation: 

Lines 109-111: "The workshop participants agreed that 109 mixed methods combining quantitative and qualitative approach is more suitable for op-110 erational research in Health ERDM, ..." - Would it be Health ERDM or Health EDRM?

The reviewer expresses his best wishes to this tough and relevant international initiative. Cheers.

Author Response

General comments

  1. It is a very good conference report that is showing its main concern of organize and develop a broad and efficient future system to enhance technical proceedings into very complex and harmful situations. This report is well structured and very clear. Only one very, very simple observation: Lines 109-111: "The workshop participants agreed that mixed methods combining quantitative and qualitative approach is more suitable for operational research in Health ERDM, ..." - Would it be Health ERDM or Health EDRM? The reviewer expresses his best wishes to this tough and relevant international initiative. Cheers.

We sincerely thank you for your careful review and for pointing out the typo. It has been corrected to EDRM. (page 3, line 123)

Reviewer 2 Report

I am grateful for the opportunity to review this manuscript. The manuscript provides an important summary of discussion that occurred among foremost disaster health leaders in Japan. Because of the intimate experience with disasters that healthcare workers and researchers have gained in this country, the insight provided by this summary of expert discussion is of great value to the wider, international disaster health research and practice community. Because of this, I believe this manuscript has great merit.

Comments:

Introduction: Very well done, giving context to the HEALTH EDRM RN, and the conference.

Materials and Methods: This section could benefit from greater detail regarding the process for summarizing discussion. What was the method used for summarizing themes? Who determined the final themes and how many researchers were involved in this process? Were themes based on information from notes, audio/video recordings? Were themes (as reported) validated in some way by the audience of experts in attendance? In summary, further clarity regarding this process would be helpful in this section to provide more rigor.

109: ‘family planning and productive health cares – please clarify what this means.

130: ‘do no harm’ principle – while I am familiar with ‘do no harm’ I don’t know exactly how this is referred to in the citation from Mary Anderson. Since this reference occurs more than once in your manuscript, it would be helpful for the reader to have an explanation of what exactly this is referring to, without forcing the reader to look up the reference.

132-141: long sentence. Break into shorter phrases for easier readability.

157-175: There are a few long sentences in this paragraph that make reading difficult. Please break down into shorter phrases.

185-188: rephrase into active voice instead of passive voice.

189: high standards?

197: subject verb agreement: should be risk reductions and related areas are lower…

206: Operational research or operational research? Please stay consistent throughout the manuscript.

224: rephrase line: replace ‘size’ with ‘magnitude’ or ‘population’?

227: is this referring to people with dementia/cognitive disabilities in emergencies and disasters in Japan?  Or in general all disaster situations (not limited to Japan)?

233-234: clarify ‘loss of function of healthcare…. systems/ services/ supplies/ workers?

239: greatly affecting

245: rephrase/clarify: It is not clear what ‘those’ refers to. Also, ‘long-shelf life food’ is not something I am accustomed to reading

My suggestion: …the diet of many people in the shelters was mostly composed of highly processed food containing a high percentage of protein and carbohydrates

252-262: this paragraph is composed of two very long sentences. Please breakdown into shorter sentences for easier reading. Also, this section is not clear what ‘nursing’ is referring to. Does this refer to nursing care and the role of nurses in disaster care? Please provide some clarity here for nurses’ role, as nurses are involved in disaster health care in many countries.

259: public health nurses?

264-266: rephrase to put verb closer to beginning of sentence for easier reading. Also, this sentence introduces community based integrated care system, which I am not familiar with. This could be explained with a little more detail or clarified in this section.

274: check if ‘are working’ is correct…. Perhaps … ‘volunteers work to provide…”

292-295: place verb earlier in sentence for easier reading.

313-317: please rephrase this sentence as there seems to be awkward phrasing throughout.

Perhaps… ‘…on the five areas highlighted in this workshop is being significant….’
Also, I’m not certain what ‘ultimate damage’ refers to
consider: ‘…is both an urgent need, and will contribute to future pandemic response.’

Author Response

General comments

  1. I am grateful for the opportunity to review this manuscript. The manuscript provides an important summary of discussion that occurred among foremost disaster health leaders in Japan. Because of the intimate experience with disasters that healthcare workers and researchers have gained in this country, the insight provided by this summary of expert discussion is of great value to the wider, international disaster health research and practice community. Because of this, I believe this manuscript has great merit.

We sincerely appreciate your careful review and thank you for pointing out the grammatical and other errors. The manuscript has been carefully proofread and edited by a native English speaker.

Our response to other major comments can be referred to the attached word file. 

Best regards,

Shuhei Nomura and Ryoma Kayano

Reviewer 3 Report

This article is a straightforward presentation of the conference proceedings, and it reads very well. My only comment is that there are some occasions where the English is a little stilted -- primarily with respect to the usage of the articles "a", "an", and "the". An editor should easily be able to spot the problems.

All in all, I found the content interesting and important.

Author Response

General comments

  1. This article is a straightforward presentation of the conference proceedings, and it reads very well. My only comment is that there are some occasions where the English is a little stilted -- primarily with respect to the usage of the articles "a", "an", and "the". An editor should easily be able to spot the problems. All in all, I found the content interesting and important.

We sincerely thank you for your careful review. The manuscript has been carefully proofread and edited by a native English speaker.

Reviewer 4 Report

The article develops an interesting framework that priorities to promote operational research to better meet the needs of emergency and and efforts to translate science to policies and programs.

To achieve this goal, authors through a workshop targeted on 20 experts from different research fields and discussed on three main topics.

 The article would be improved if authors, in addition to what they reported, also presented some more elements of corresponding applied cases to translate design and implementation of operational research into policy. Furthermore, additional literature would strengthen the quality of the article.

In the Introduction section, a Table showing the information presented would help.

In general, it is an interesting and well-written article which from the qualitative point of view in which it moves, raises significant and substantiated questions. However, I believe it needs to be enriched with potential applications (specific cases) concerning the scientific area it presents.

Author Response

General comments

  1. The article develops an interesting framework that priorities to promote operational research to better meet the needs of emergency and efforts to translate science to policies and programs. To achieve this goal, authors through a workshop targeted on 20 experts from different research fields and discussed on three main topics.

Our response to other major comments can be referred to the attached word document. 

We sincerely appreciate your careful review.

Best regards,

Shuhei Nomura and Ryoma Kayano
